# Diabetes-related distress and associated factors among adult diabetes mellitus patients attending public hospitals in Gedio zone, southern Ethiopia: Mediation analysis

Alem Bayable Mersha[ID][1]*, Atirsaw Shimekaw[1], Edom Gebremedhin[2], Jerusalem Sewalem[3], Andualem Bayih[4], Abel Desalegn Demeke[1], Simret Gebre[3]

1 Department of Nursing, College of Medicine and Health Science, Dilla University, Dilla, Ethiopia, 2 Department of Internal Medicine, College of Medicine and Health Science, Dilla University, Dilla, Ethiopia, 3 Department of Psychiatry, College of Medicine and Health Science, Dilla University, Dilla, Ethiopia, 4 Department of Medical Laboratory, College of Medicine and Health Science, Dilla University, Dilla, Ethiopia

* alembayable122@gmail.com

## Abstract

### Background

Individuals with diabetes might experience distress due to their treatment regimen, interactions with physicians, interpersonal relationships, and emotional well-being. This may lead to lower self-efficacy, which would impair self-treatment adherence and result in poor glycemic control. Understanding and addressing gaps may help individuals with diabetes improve their glycemic control. Therefore, this study aimed to determine the magnitude of diabetes-related distress and associated factors among adult diabetes mellitus patients attending public hospitals in the Gedio zone, Southern Ethiopia.

### Method

An institutional-based cross-sectional study was conducted from April to May 2024 at four public hospitals in the Gedio zone. Study participants who met the inclusion criteria were selected using systematic random sampling techniques. Data were collected using the Diabetes Distress Scale (DDS-17). Both bivariable and multivariable ordinal logistic regression were applied. In multi-variable ordinal logistic regression, a P value < 0.05 significant level was used to identify factors of diabetic-related distress. Mediation analyses were conducted using the bootstrapping technique. A variable showing a significant indirect effect was considered as a mediator.

**Data availability statement:** All relevant data are within the paper and its Supporting information files.

**Funding:** The research project was conducted from April to May 2024, and it was fully sponsored by Dilla University, but it had no role in study design, data collection and analysis, decision to publish, or preparation of the manuscript.

**Competing interests:** The authors have declared that no competing interests exist.

## Results

In this study, a total of 506 adult patients with diabetes participated. The overall prevalence for moderate and high levels of diabetes-related distress was 47.04% (95% CI;42.6–51.5) and 41.5% (95% CI; 37.2–45.9), respectively. Variables like never had planned physical activity (adjusted odd ratio, aOR=3.09;95%CI = 1.034–9.28; P = 0.04), poor social support (aOR=3.44;95%CI = 1.42–8.33;P < 0.006), complications (aOR=1.79; 95%CI = 1.13–2.84; P < 0.013) and high blood pressure (aOR= 1.85; 95%CI = 1.12–3.05; P = 0.016) were factors for diabetes related distress. Depression was identified as a partial mediator of the relation between social support and diabetes related distress.

## Conclusion

Diabetes-related distress was highly prevalent in diabetes patients. Healthcare providers need to address this by integrating psycho-social care with collaborative medical care.

## Introduction

Diabetes mellitus is a group of long-term metabolic problems indicated by abnormalities in insulin secretion, action, or both [1]. It is a leading cause of morbidity and mortality among people all over the world, and is associated with an increase in unhealthy lifestyles, such as poor eating habits and insufficient physical activity [2–4]. Individuals with long-term diabetes may face not only physical health challenges but also experience negative psychological issues because of the risk of complications and social burden that comes with the disease, which can result in an emotional burden. As a result, managing DM and achieving the desired targeted glycemic control requires additional duties, planning, and self-monitoring [5,6].

The term "diabetes-related distress" (DRD) refers to the worries, concerns, and emotional problems that are associated with managing diabetes, getting support, bearing emotional burdens, and getting access to care [7,8]. When people with diabetes believe that their coping mechanisms are inadequate to manage the threat of their illness, this leads to an emotional burden that is specific to diabetes. Diabetic distress has been divided into four domains, which include emotional, interpersonal, physician, and regimen-related distress [9]. Patients with high DRD are likely to demonstrate poor self-management due to the distress of daily needs of disease management, worries about poor glycemic control, fears about diabetic complications, inadequate support from significant others, stigma, and financial struggles [10,11]. It is widely known that self-care practice and medication adherence determine glycemic control, which is connected to future problems and a lower quality of life [12]. Psychological conditions associated with diabetes can create hypothetically dangerous surroundings that can influence health-seeking behavior, acceptance of the diagnosis, and treatment adherence, which have a significant impact

on patients health outcomes. Individuals with DRD are 1.56 times more likely to mortality, and higher risk for morbidity [13]. Different Studies found diabetes distress was significantly correlated to the number of serious complications, including heart disease, stroke, blindness, kidney failure, and lower-limb amputation due to uncontrolled glycemia [2,14–16]. It can also lead to severe medical and psychological outcomes, including reduced physical activity [17], unhealthy eating practices [18], less frequent self-monitoring of blood glucose [19,20], elevated HbA1c [21], more frequent severe hypoglycemia [22], and lowered quality of life [23].

Despite Ethiopia being one of the four nations in sub-Saharan Africa with the highest rates of adult diabetes, there are limited studies conducted on the psychological aspect of diabetes in the country [24]. This calls for a change in healthcare provider systems that manage diabetic patients by taking psychological aspects like diabetes-related distress into account [25]. So, this study intended to fill gaps by conducting scientific research on this topic to rule out the magnitude and factors that could affect diabetes-related distress by including individuals with type 1 and type 2 diabetes mellitus.

## Materials and methods

### Study area

The study was conducted at public hospitals in the Gedio zone. Gedio zone is located 359 kilometers south of Addis Ababa and roughly 90 kilometers from Hawassa (capital city of Sidama region). According to the Central Statistical Agency report, the zone has a total population of 847,434 [26]. It has twelve districts or woredas, including Dilla Town, Kochore, Chelelektu Town, Gedeb Town, Chorso, Yirga Chefe Town, Yirga Chefe woreda, Dilla Zuria, Gedeb woreda, Wonago, Bule, and Raphe. There are four public hospitals in the zone, which include one general hospital in Dilla town and three primary hospitals in Bule, Gedeb, and Yirga Chefe weredas.

### Study design and period

An institution-based cross-sectional study was conducted among all diabetic patients who had follow-ups at the outpatient departments of public hospitals in the Gedeo Zone, Southern Ethiopia, from April 20 to May 20, 2024.

### Population

All patients with diabetes mellitus who were receiving follow-up care at the outpatient department of public hospitals in the Gedio zone were considered the source and study population.

### Inclusion and exclusion criteria

This study included adult patients aged 18 years or older who had been diagnosed with either type 1 or type 2 diabetes mellitus and had at least three months of follow-up up but patients who were unable to communicate due to their critical health condition, had a history of psychiatric disorders, and newly diagnosed patients with type 1 or type 2 diabetes were excluded from the study.

### Sample size determination and sampling technique

The sample size was calculated using Epi Info version 7.2. The total sample size was calculated with a 95% confidence level, 80% power, and a 1:1 ratio of unexposed to exposed groups using the double population-proportion formula. The sample size was determined by using the previous study variable of HbA1c (percentage of outcome in the unexposed group = 93.2%, and adjusted odds ratio = 5.49) [27]. So, after adding a 10% non-response rate, the final sample size for this particular study was 521.

The list of respondents, which serves as a sampling frame, was obtained from the follow-up clinic registration logbooks of four public hospitals in the Gedio Zone. After establishing the sampling frames in each hospital, the total sample size

was first proportionally allocated according to the number of registered diabetes mellitus patients in each hospital to facilitate the selection of study participants. Then, a systematic random sampling technique with a sampling interval of three was applied to identify the study unit to be included in the study. The people living with diabetes who met the inclusion criteria were recruited for the study until the required sample size was achieved. After proportional allocation, the number of participants recruited from each hospital was 227 from Dilla University Teaching Hospital, 144 from Yergachefe Primary Hospital, 78 from Gedeb Primary Hospital, and 72 from Bule Primary Hospital.

## Operational definition

**Diabetic-related distress (DRD)** is defined as the concerns, worries, and emotional burdens that are negative emotional experiences when managing diabetes. An overall mean score of less than 2 is regarded as little distress, a score ranging from 2 to 2.9 is considered moderate distress, and a score of 3 or higher is considered a high level of distress [9,28].

 **Diabetic mellitus (DM)** was defined in this study as a metabolic disorder characterized by the presence of high glucose levels due to poor insulin secretion (Type-1) or poor insulin utilization (Type-2) [29]. In this study, information on the type of diabetes was obtained from patients medical records. The criteria used by each hospital to define poor glycemic control were based on fasting blood glucose (FBG) levels of < 70 mg/dL or ≥ 130 mg/dL.

 **Social support** was assessed using the Oslo 3-item social support scale, which assesses the level of support the patient received from family and friends. This scale score ranged from 3 to 14, with scores of 3–8 indicating poor support, 9–11 shows moderate support, and 12–14 was considered as high support [30].

 **Substance use** was assessed by using alcohol, smoking, and substance involvement screening test (ASSIST) questionnaires which classified as low risk (0–10 for Alcohol intake and 0–3 for other substances), moderate risk (11–26 for alcohol intake, and 4–26 for other substances), and high risk (greater than 27 for all substance used) [31,32].

 **Depression** is a mental disorder characterized by sadness and lack of interest or pleasure in previously rewarding or enjoyable activities, assessed using PHQ-9. A PHQ score of 4–9, 10–19, and greater than 20 suggests mild, moderate, and severe depression, respectively [33].

## Data collection tools

Data was collected using a prepared, structured, and pretested questionnaire developed by reviewing different relevant Literature [34–36] that can address the objective of the study. DDS-17 is a well-validated and widely used tool designed to measure various diabetes related [37–39].Each question has six answer choices: 1(no problem), 2 (slight problem), 3(moderate problem), 4(somewhat serious problem), 5 (a serious problem), and 6 (a very serious problem). The questionnaire contains four domains: emotional burden (contain 5 items: questions 1, 3, 8, 11, and 14); physician-related distress (contain 4 items: questions 2, 4, 9, and 15); regimen-related distress (contain 5 items: questions 5, 6, 10, 12, and 16); and interpersonal related distress (contain 3 items: questions 7, 13, and 17). An overall mean score of less than 2.0 was considered as little to no distress, a score of 2–2.9 as moderate distress, and a score greater than 3 was considered a high level of distress. A four-point Likert scale of PHQ-9 and three items of Oslo social support were used to assess depression and participants level of social support, respectively [30,33]. In addition to this, substance use was assessed using a standardized alcohol, smoking, and substance involvement screening test (ASSIST). The questionnaires for this study consist of four sections (section one: socio-demographic characteristics, section two: diabetes-related distress, section three**:** clinical-related characteristics, and section four: patient-related characteristics). These sections of the questionnaire were either administered through face-to-face interviews for patients who could not read or write, or completed by the patients themselves. Additionally, some clinical data, such as type of diabetes, fasting blood glucose level (FBG), and blood pressure level, were obtained by reviewing patients charts.

## Data collection methods

Data was collected by eight trained BSc degree nurses through face-to-face interviews, self-administered questionnaires, and reviewing the patient charts, and the whole activity of data collection was followed by four supervisors. Written permission to conduct the study was obtained from each hospital before the recruitment of participants for the study. Patients were informed about the study's purpose and provided written consent before participation. All data collection activities were carried out following patient consent to participate in the study. Interviews were carried out in a quiet room during routine follow-up visits, and participants were approached after completing their regular medical care to ensure minimal disruption.

## Data quality assurance

The questionnaire was first prepared in English, translated into the local language (Gedioffa and Amharic) by a bilingual translator, and then back-translated into English by another bilingual translator to ensure consistency. The validation of the data collection tool for local languages was checked by incorporating two experts in the field to assess the translated questionnaire for face validity. In addition, to ensure the questionnaires clarity and ease of use, a pretest was conducted in 26 (5% of the total sample size) adult diabetes patients at DUTH. These participants were excluded from the final analysis. Based on the pretest results and experts feedback, minor modifications were made to the questionnaire to enhance clarity and consistency of wording before actual data collection began. The reliability (internal consistency) of the instruments was determined using Cronbach's alpha coefficient and found to be 0.87 for the general diabetic distress scale,0.83 for depression, 0.79 for substance use, and 0.67 for the social support scale. Before the actual data collection, the principal investigator gave two days of training to the data collectors and supervisors about the objectives of the study, questionnaires, and data collection techniques. Each questionnaire was reviewed to ensure completeness, accuracy, and consistency of collected data.

## Data processing and analysis

After the data collection, the completeness and consistency of the questionnaires were checked manually, and before analysis, missing values were checked by the principal investigator. Then both the questionnaires and the variables are coded, categorized, and entered into Epi Data version 7.2 and exported to SPSS version 25 for cleaning and further analysis. The analysis was supported using descriptive interpretation, frequency tables, and summary statistics (mean and median) to summarize the socio-demographic, clinical, and patient characteristics of the study participants. Multi-collinearity between independent variables was checked using the variance inflation factor (VIF) and tolerance. The values of both statistics for all independent variables were $VIF < 10$ and $tolerance > 0.1$. In this study, the proportional odds assumptions were checked with the test of the parallel line, which indicates a p-value of 0.587. Ordinal logistic regression was carried out using a generalized linear model in SPSS to identify the association between each variable with the outcome variable. Variables with a $p-value < 0.25$ in the bivariable ordinal logistic regression became candidates for multivariable ordinal logistic regression. Then, factors that have a $p-value < 0.05$ were considered predictors of diabetic-related distress. The results were expressed in terms of crude and adjusted odds ratios with 95% confidence intervals. A $p-value < 0.05$ was used to determine statistically significant predictors of diabetes related distress.

Mediation analysis was conducted on diabetic patients to explore the causal pathway from the independent variables to the dependent variable through mediating variables. In this study, depression was identified as the mediating variable, as it was significantly associated with both the outcome variable (diabetes-related distress) and the predictor variable (social support). The analysis assessed several components of the mediation model, including the total effect of the social support on the diabetic related distress (path c), the direct effect of the social support on depression (path a), the effect of depression on diabetic related distress while controlling for the social support (path b), and the direct influence of social

support on diabetic related distress after controlling for the effect of depression (path c′). The indirect impact was estimated using the product of paths a and b. This component of mediation analysis was easily tested using the PROCESS macro-model 4 in SPSS version 25. Additionally, the percentage of the effect attributed by the mediator was determined by dividing the indirect effect by the total effect (the sum of the indirect and direct effects), followed by multiplying by 100%. All results were reported in tabular form with regression coefficients, p-values, and 95% confidence intervals, based on bootstrap estimation.

### Ethical consideration

The study was carried out after a letter of approval was obtained from the ethical review committee (protocol unique number: duirb/045/23–06) of Dilla University College of Medicine and Health Science, which also facilitated an official letter to be written to the selected hospitals to get their permission and cooperation for the study. Approval was also obtained from participating hospitals. Written informed, voluntary consent was taken from each study participant who was selected for the study. Before the interview begins, the data collector assures participants that the information they give is used only for research purposes and kept anonymous throughout data collection.

## Results

### Socio-demographic characteristics of diabetic patients

A total of 506 diabetic patients fulfilled the inclusion criteria and participated. Among these, 55.1% (279) of males and 44.9% (227) of females participated in this study. The mean age of the patients was 51.12 ± 16.6 years, while the maximum and minimum ages of patients visiting the outpatient department were 88 and 18 years, respectively. In terms of marital status, 54.5% (276) were married. Additionally, about 19.4%(98) of the sample had not received any formal education (Table 1).

### The magnitude of diabetic-related distress among diabetic patients

This study found that 88.54% of diabetes patients had diabetic-related distress, in which 41.5% (95%CI; 37.2–45.9) of patients had high distress and 47.04% (95%CI;42.6–51.5) had moderate distress (Fig 1).

### Clinical-related characteristics of diabetic patients

The majority, 378(74.7%) of diabetic patients had been living with diabetes for less than 7 years, and 205(40.5%) patients had experienced diabetic-related complications; among these, 84(16.6%) developed diabetes foot ulcers. Regarding diabetic medications, 266 (52.6%) of respondents were taking both oral medication and insulin. The study also revealed that 411(81.2%) of the study participants had FBG levels of ≥ 130 mg/dl, and more than half, 379(74.9%), were diagnosed with type 2 diabetes (Table 2).

### Patient-related characteristics of diabetic patients

Among 506 patients attending the outpatient department, 245(39.0%) had never engaged in planned physical activity, and 446 (88.1%) had reported an average sleep duration of less than eight hours. More than half, 326(64.4%) of the patients experience poor social support (Table 3).

### Mediation analysis

This study conducted a mediation analysis to examine whether depression serves as a mediator in the relationship between social support and diabetes-related distress among diabetic patients. The total effect of social support on diabetic distress was statistically significant ($\beta = -0.0635$, 95% CI [- 0.10, −0.026], $P < 0.001$), indicating that higher social

**Table 1. Socio-demographic characteristics of adult diabetes mellitus patients.**

| Variables | Categories | Frequency (%) |
|---|---|---|
| Sex | Male | 279(55.1) |
| | Female | 227(44.9) |
| Age | <52yrs | 230(45.5) |
| | ≥52yrs | 276(54.5) |
| | Mean±SD | 51.12±16.6 |
| Marital status | Single | 106(20.9) |
| | Married | 276(54.5) |
| | Widow | 69 (13.6) |
| | Divorce/Separated | 55 (10.9) |
| Educational level | No formal education | 98 (19.4) |
| | Able to read and write | 134(26.5) |
| | Primary Education | 125(24.7) |
| | Secondary Education and above | 149 (29.4) |
| Occupation status | Unemployed | 167(33.0) |
| | Governmental Employed | 104(20.6) |
| | Private Employed | 48(9.5) |
| | Farmer | 68(13.4) |
| | Merchant | 119(23.5) |
| Place of residence | Urban | 348(68.8) |
| | Rural | 158(31.2) |

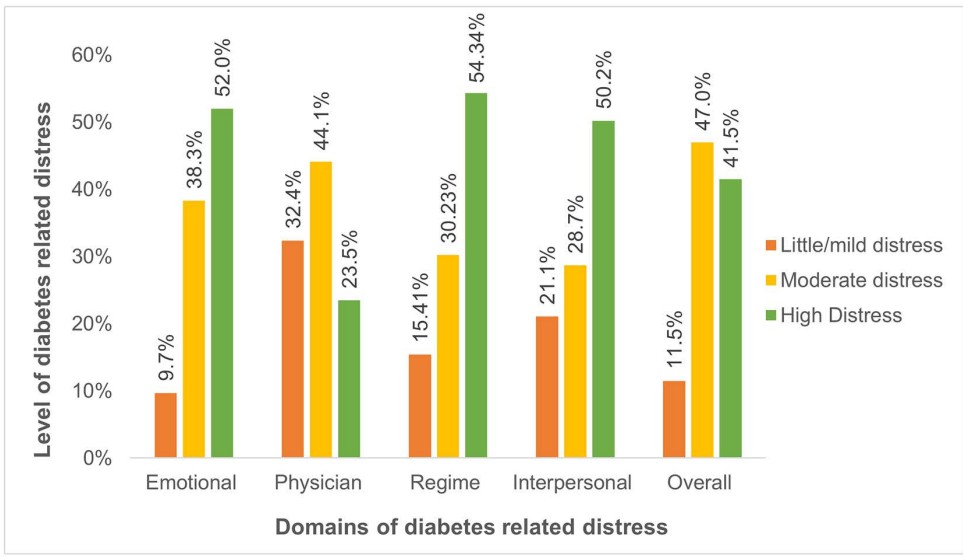

**Fig 1. Prevalence of diabetic-related distress and its domains among adult diabetes mellitus patients.**

support is associated with lower levels of diabetic distress. When depression was included as a mediator in the model, the direct effect of social support on diabetic distress remained statistically significant ($\beta$=−0.056,95% CI [- 0.0924, −0.0189], $P$=0.003). This suggests that social support continues to have a direct impact on diabetic distress even after considering

**Table 2. Clinical-related characteristics of adult diabetes mellitus patients.**

| Variables | Categories | Frequency (%) |
|---|---|---|
| Duration of living with diabetes | ≤ 7 years | 378(74.7) |
| | 8-14years | 100(19.8) |
| | ≥15years | 28(5.5) |
| | Median [IQR] | 6 [5–7] |
| Diabetic associated complications | Yes | 205(40.5) |
| | No | 301(59.5) |
| Type of complication | Nephropathy | 32(6.3) |
| | Retinopathy | 62(12.3) |
| | Diabetic foot ulcer | 84(16.6) |
| | Diabetes related Coronary disease | 27(5.3) |
| Hypoglycemic in the last 3 months | Yes | 102 (20.2) |
| | No | 404(79.8) |
| Type of treatment | Oral | 188(37.2) |
| | Insulin | 52 (10.3) |
| | Oral and insulin | 266 (52.6) |
| Depression | Mild (4–9) | 49(9.7) |
| | Moderate (10–19) | 213(42.1) |
| | Sever (≥20) | 244(48.2) |
| Fast blood glucose (FBG mg/dl) | 70-99 mg/dl | 39(7.7) |
| | 100-129 mg/dl | 56(11.1) |
| | ≥130 | 411(81.2) |
| Blood pressure (mm Hg) | Normal (<120/80) | 97(19.2) |
| | Pre-hypertension (120–139/80–89) | 196(38.7) |
| | Hypertension (≥140/90) | 213(42.1) |
| BMI (kg/m) | Normal weight (18.5–24.9 kg/m$^2$) | 16(3.2) |
| | Overweight (25–29.9 kg/m$^2$) | 158(31.2) |
| | Obese (≥30 kg/m$^2$) | 332(65.6) |
| Type of DM | Type 1 | 127(25.1) |
| | Type 2 | 379(74.9) |

the mediating role of depression. Furthermore, the indirect effect of social support on diabetic distress through depression was also statistically significant ($\beta = -0.0078$, 95% CI [–0.0162, –0.0015]). This all implies that depression partially mediates and explains 12.28% of the social support effect on diabetic distress.

### Factors associated with diabetic-related distress among diabetes patients

Out of twenty-one independent variables, thirteen variables, namely age, educational status, marital status, planned exercise, treatment type, duration of diabetes, blood pressure, social support, complications, hypoglycemia, FBG, depression, and BMI, showed an association in bivariable analysis at p value < 0.25. These variables were then entered into the multivariable logistic regression analysis. Planned exercise, blood pressure status, social support, and complications were identified as statistically significant predictors of diabetic-related distress among diabetes patients with a p-value < 0.05.

Our study shows that patients who have been physically inactive or who never had planned physical activity were approximately three times more likely to have diabetic distress (aOR= 3.09; 95%CI = 1.034–9.28; P = 0.04). The chance that a person who has diabetic complications was 1.79 times more likely to have diabetic-related distress than a person

**Table 3. Patient-related characteristics of adult diabetes mellitus patients.**

| Variables | Categories | Frequency (%) |
|---|---|---|
| Planned physical exercise program | Never | 469(92.7) |
| | Once or twice a week | 22(4.3) |
| | Regular exercise (≥ three times a week) | 15(3.0) |
| Average Duration of Sleep (in hours) | <8hrs | 446 (88.1) |
| | ≥8hrs | 60 (11.9) |
| | Mean ±SD | 6.28±1.15 |
| Alcohol intake | Low risk | 374(73.9) |
| | Moderate risk | 95(18.8) |
| | High risk | 37(7.3) |
| Cigarette smoking | Low risk | 343 (67.8) |
| | Moderate risk | 128(25.3) |
| | High risk | 35(6.9) |
| Social support | Poor support | 326(64.4) |
| | Medium support | 154(30.4) |
| | High support | 26(5.1) |

who did not have complications (aOR=1.79; 95%CI = 1.13–2.84; P < 0.009). In addition, patients with high blood pressure are 1.8 times more likely to experience distress than normal patients (aOR=1.85; 95%CI = 1.12–3.05; P = 0.016). Finally, patients who had lower social support were 3.4 times more likely to have diabetic-related distress than higher social support (aOR=3.44; 95%CI = 1.42–8.33; P < 0.006) (Table 4).

## Discussion

This study was conducted to assess the level of DRD and its associated factors among adult diabetes mellitus patients attending public hospitals in Gedio zone, southern Ethiopia. The study showed that 11.46% (95%CI;8.8–14.6) had little or mild distress, while 47.04% (95%CI;42.6–51.5) and 41.5% (95%CI; 37.2–45.9) had moderate and high levels of diabetic-related distress, respectively. The findings of this study were consistent with the study conducted in Amhara regional state [40], which reported that 43.2% of patients had a moderate level of distress and 44.4% experienced a high level of diabetic distress. Similar results were also observed in southeast Ethiopia [17], in which 41.8% of the participants suffer from a higher level of DRD. However, it is higher in comparison with previous studies conducted in Vietnam (23.6%) [41], and Barbadian(17%) [42]. The higher prevalence in this study could be due to poor quality of diabetes care service, and the measurement tool used to quantify the level of diabetic distress. The highest level of distress was observed in the Regime-related domain of diabetic-related distress, which is in line with the study done in Iran [34]. The possible explanation is that decreased medication adherence has an impact on glycemic control, which results in further complications and decreased self-management [43]. To reduce the impact of regimen-related distress resulting from poor glycemic control due to medication non-adherence, the application of cognitive-behavioral therapy may lead to significant improvements in HbA1c levels and help to improve blood glucose levels, thereby reducing distress [44]. In addition, diabetes health-coaching programs are beneficial in improving glycemic control and reducing distress [45]. The lowest level of physician-related distress in our study may be due to the patients higher satisfaction with the doctor-patient relationship [34].

Our study shows that physically inactive participants were more likely to have diabetes distress. This finding is consistence with the conclusions drawn from previous research carried out in the Amhara region and Saudi Arabia. The potential reason could be that those who did not engage in regular physical activities might think they do not sufficiently adhere to their supportive self-care management, which can lead to high regimen-related distress [40,17,46]. Patients

**Table 4. Bivariable and multivariable ordinal logistic regression analysis for factors associated with diabetic-related distress.**

| Variables | Categories | DRD | | | OR (95%CI; P value) | aOR (95%CI) | P value |
|---|---|---|---|---|---|---|---|
| | | Mild | Moderate | Severe | | | |
| Age | <52yrs | 30 | 111 | 89 | 0.79 (0.57 - 1.1; 0.18) * | 1.0(0.65-1.55) | 0.98 |
| | ≥52yrs | 28 | 127 | 121 | Reference | Reference | |
| Educational status | Uneducated | 16 | 48 | 34 | 0.72(0.44 −1.17; 0.19) * | 0.63(0.37-1.07) | 0.087 |
| | Able to read and write | 13 | 57 | 64 | 1.28(0.81-2.0; 0.28) | 1.07(0.64-1.79) | 0.80 |
| | Primary Education | 13 | 61 | 51 | 1.0(0.64-1.57; 0.99) | 0.94(0.58 −1.54) | 0.81 |
| | Secondary Education and above | 16 | 72 | 61 | Reference | Reference | |
| Marital status | Single | 12 | 49 | 45 | 0.95(0.62-1.45; 0.80) | 1.31(0.76-2.28) | 0.33 |
| | Married | 20 | 142 | 115 | Reference | Reference | |
| | Widow | 15 | 25 | 28 | 0.7(0.41-1.19; 0.23) * | 0.59(0.33-1.06) | 0.079 |
| | Divorce/Separated | 11 | 22 | 22 | 0.71(0.40-1.26; 0.18) * | 0.64(0.34-1.19) | 0.16 |
| Planned Exercise | Never | 49 | 220 | 200 | 2.5(0.87-7.21; 0.069) * | 3.09(1.03-9.28) ** | **0.043** |
| | Once or twice a week | 4 | 13 | 5 | 1.13(0.30 - 4.17; 0.847) | 1.32(0.34 −5.05) | 0.68 |
| | Regular exercise (≥ three times a week) | 5 | 5 | 5 | Reference | Reference | |
| Treatment type | Oral | 25 | 82 | 81 | 0.93(0.65- 1.34; 0.71) | 1.13 (0.75-1.7) | 0.55 |
| | Insulin | 10 | 25 | 17 | 0.58(0.33-1.03;0.06) * | 0.69 (0.35-1.36) | 0.29 |
| | Oral and insulin | 23 | 131 | 112 | Reference | Reference | |
| Duration of DM | ≤ 7 years | 46 | 185 | 147 | Reference | Reference | |
| | 8-14years | 7 | 44 | 49 | 1.54(1.0-2.36; 0.04) * | 1.32(0.79-2.19) | 0.28 |
| | ≥15years | 5 | 9 | 14 | 1.28(0.59 - 2.7; 0.50) | 0.93(0.40-2.14) | 0.87 |
| Blood pressure status | Normal | 20 | 47 | 30 | Reference | Reference | |
| | Pre-hypertension | 22 | 94 | 97 | 2.05(1.28-3.27; 0.006) * | 1.64(0.97-2.76) | 0.063 |
| | Hypertension | 16 | 97 | 83 | 1.93(1.20- 3.09; 0.002) * | 1.85(1.12- 3.05) ** | **0.016** |
| Social support | Poor support | 10 | 72 | 72 | 5.08(2.19 - 11.8; <0.001) * | 3.44(1.42-8.34) ** | **0.006** |
| | Medium support | 37 | 157 | 131 | 3.72(1.65 −8.37; 0.001) * | 2.29(0.96-5.46) | 0.061 |
| | High support | 11 | 9 | 7 | Reference | Reference | |
| Complications | Yes | 12 | 96 | 97 | 1.67(1.19 −2.36; 0.003) * | 1.79(1.13-2.84) ** | **0.013** |
| | No | 46 | 142 | 113 | Reference | Reference | |
| Hypoglycemia | Yes | 13 | 52 | 37 | 0.78(0.52- 1.18; 0.24) * | 0.64(0.4-1.02) | 0.062 |
| | No | 45 | 186 | 173 | Reference | Reference | |
| Fast blood glucose level | 70-99 mg/dl | 10 | 15 | 14 | Reference | Reference | |
| | 100-129 mg/dl | 8 | 27 | 21 | 1.41(0.63-3.16; 0.39) | 0.98(0.413- 2.36) | 0.97 |
| | ≥130 | 40 | 196 | 175 | 1.83(0.95-3.54; 0.06) * | 1.29(0.64-2.60) | 0.48 |
| Depression | Mild (4–9) | 11 | 19 | 19 | Reference | Reference | |
| | Moderate (10–19) | 33 | 111 | 69 | 0.97 (0.52 −1.8; 0.93) | 1.66(0.86-3.187) | 0.13 |
| | Sever (≥20) | 14 | 108 | 122 | 2.13 (1.15 −3.93; 0.01) * | 0.80(0.42-1.53) | 0.50 |
| BMI | Normal weight (18.5–24.9 kg/m$^2$) | 34 | 156 | 142 | Reference | Reference | |
| | Overweight (25–29.9 kg/m$^2$) | 19 | 77 | 62 | 0.86 (0.59-1.23; 0.41) | 0.82(0.55 - 1.22) | 0.34 |
| | Obese (≥30 kg/m$^2$) | 5 | 5 | 6 | 0.49 (0.17-1.37, 0.14) * | 0.76(0.25-2.35) | 0.64 |

**Note**; OR= crude odd ratio; aOR= adjusted odd ratio, *=p Value <0.25; **=p Value <0.05.

who have complications are significantly associated with higher levels of distress, which is in line with previous studies done in southwest Ethiopia [47]. People living with diabetes may become emotionally burdened, frustrated, discouraged by the threat of developing complications, and perceive themselves as helpless to avert them. This causes distress

and results in withdrawal from their self-care routines [17]. This study also found that people with high blood pressure are more likely to have DRD. The possible explanation for this finding could be the result of experienced side effects of antihypertensive medication [48]. Additionally, the odds of DRD increase in patients who had low social support, which is in line with studies done in Indonesia and southeast Ethiopia [36,17]. The possible reasons for this could be social support from family or friends, as a form of emotional, informational, or financial can help the patient cope with problems and give emotional strength. Furthermore, this study shows that a lower level of social support leads to higher diabetic related distress through the mediating variable of depression. Previous studies have separately demonstrated that poor social support is associated with increased levels of both depression [49] and diabetic related distress [50], and that depression is positively correlated with distress [51,52]. This may be explained by the fact that individuals without adequate social support may first experience feelings of loneliness, helplessness, and hopelessness. These negative behaviors increase or worsen diabetic related stressors, emotional burden, frustration, and make them more vulnerable to distress.

## Conclusion and recommendations

Generally, this study reveals that 47.1% of diabetic patients experience a moderate diabetic related distress, while 41.5% face a high level of diabetic related distress. The results further suggest that complications, high blood pressure, inadequate social support, and absence of planned physical exercise are important factors that contribute to increased levels of distress in diabetic patients. This study also indicates that depression has a significant mediation role between social support and diabetes related distress.

Healthcare facilities should implement targeted strategies to reduce diabetes-related distress by addressing the key contributing factors identified in this study. This includes integrating complication monitoring into routine follow-up visits, which enables healthcare providers to regularly screen for complications and offer timely interventions to ease both the physical and emotional burdens on patients. Additionally, giving training for social health workers to assess patients social support status and link them to counseling can strengthen family and social connections, thereby helping to reduce distress. Regarding depression, due to its significant effect on social support and diabetic distress as a mediator, healthcare providers should plan to incorporate routine screening and effective management of depression into patient care. Furthermore, addressing comorbid conditions such as hypertension and promoting emotional well-being should be integrated into routine care. Moreover, healthcare providers are encouraged to offer individualized exercise counseling to help patients establish and maintain a regular physical activity routine. Finally, we recommend that future researchers should explore longitudinal and intervention-based designs to better understand the causal pathway and to test the effectiveness of a specific intervention in reducing DRD.

## Limitations of the study

Despite this study providing meaningful contributions, it has some limitations that should be considered when interpreting the results. First, the cross-sectional nature of the study only determines the association between variables, not the causality or direction. Therefore, future experimental studies are required to identify the causality between variables. Second, the data collected is based on self-reporting and might be subject to recall and response bias, which might influence the result. Participants in this study were selected from hospital patients, which may introduce selection bias because people in the hospital may have severe DM, and individuals who can manage their condition well at home are often missing from the sample. Thirdly, it is difficult to determine whether the distress occurred before the onset of diabetes or is entirely associated with the disease. Finally, some participants stated that they were in a hurry to get to work and did not have time due to their busy schedules. As a result, they were unwilling to complete the questionnaire and were recorded as non-respondents. In the future, if a participant is unable to make time for an in-person interview, it is better to ask the participant to complete the questionnaire through a phone call or email.

## Supporting information

**S1 Questionnaires.** English version questionnaire for diabetic-related distress and associated factors among adult diabetes mellitus patients attending public hospitals in Gedio zone, southern Ethiopia.
(DOCX)

**S2 Questionnaires.** Amharic version questionnaire for diabetic-related distress and associated factors among adult diabetes mellitus patients attending public hospitals in Gedio zone, southern Ethiopia.
(DOCX)

**S3 Questionnaires.** Gedioffa version questionnaire for diabetic-related distress and associated factors among adult diabetes mellitus patients attending public hospitals in Gedio zone, southern Ethiopia.
(DOCX)

**S4 Data.** Overall data on diabetic-related distress and associated factors among adult diabetes mellitus patients attending public hospitals in Gedio zone, southern Ethiopia.
(XLSX)

## Acknowledgments

We extend our sincere thanks to the study participants from the public hospitals in the Gedio zone who provided us with the essential information. We also express our heartfelt gratitude to the data collectors and supervisors for their time and dedicated efforts.

## Author contributions

**Conceptualization:** Alem Bayable Mersha, Atirsaw Shimekaw.

**Data curation:** Alem Bayable Mersha.

**Formal analysis:** Alem Bayable Mersha, Edom Gebremedhin, Simret Gebre.

**Investigation:** Alem Bayable Mersha.

**Methodology:** Alem Bayable Mersha, Atirsaw Shimekaw, Jerusalem Sewalem.

**Project administration:** Alem Bayable Mersha, Edom Gebremedhin.

**Software:** Alem Bayable Mersha, Andualem Bayih.

**Supervision:** Alem Bayable Mersha.

**Writing – original draft:** Alem Bayable Mersha, Andualem Bayih.

**Writing – review & editing:** Alem Bayable Mersha, Abel Desalegn Demeke, Simret Gebre.

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
