## [Decision Letter · Decision Letter 0]

3 Jul 2025

Dear Dr. Mersha,

Thank you for submitting your manuscript to PLOS ONE. After careful consideration, we feel that it has merit but does not fully meet PLOS ONE’s publication criteria as it currently stands. Therefore, we invite you to submit a revised version of the manuscript that addresses the points raised during the review process.

Please explain 

1-more about the validity and reliability ( psychometry )  of employed questionnaires in your study in   local language

2- This study needs  mediation analysis for  robust conclusion

We look forward to receiving your revised manuscript.

Kind regards,

Hamid Reza Baradaran, M.D., Ph.D.,

Academic Editor

PLOS ONE

Journal Requirements:

“Dilla Univeristy fund this study”

4. Please remove all personal information, ensure that the data shared are in accordance with participant consent, and re-upload a fully anonymized data set.

Additional Editor Comments:

Please explain

1-more about the validity and reliability ( psychometry ) of employed questionnaires in your study in local language

2- This study needs mediation analysis for robust conclusion

Reviewers' comments:

Reviewer's Responses to Questions

**Comments to the Author**

1. Is the manuscript technically sound, and do the data support the conclusions?

Reviewer #1: Partly

Reviewer #2: Yes

2. Has the statistical analysis been performed appropriately and rigorously?

Reviewer #1: Yes

Reviewer #2: Yes

3. Have the authors made all data underlying the findings in their manuscript fully available?

Reviewer #1: Yes

Reviewer #2: No

4. Is the manuscript presented in an intelligible fashion and written in standard English?

Reviewer #1: No

Reviewer #2: Yes

Reviewer #1: Title: Diabetic-related distress and associated factors among adult diabetes mellitus patients

attending public hospitals in Gedio zone, Southern Ethiopia

Thank you for the opportunity to review this manuscript.

• I respectfully recommend a thorough review of the manuscript for language and grammar to improve clarity and readability.

• Please specify the type of diabetes in the title. At the end of the introduction, the authors mention “like type 1 diabetes mellitus,” but there is no reference to the type of diabetes in the methods section.

• The authors wrote: “The sample size was determined by using the previous study variable of HbA1c (percentage of outcome in the unexposed group = 93.2% and adjusted odds ratio = 5.49).” Please add a reference to this section.

• Please explain the phrase “fulfilled the inclusion criteria.”

• Reference the validity and reliability of the questionnaires used in the target population.

• Mention how physical activity was assessed.

• The researchers wrote: “Data was collected by eight BSc degree nurses through face-to-face interviews, self-administered questionnaires, and reviewing the patient charts depending on the occasion.” Please specify which questionnaire was collected using each of the mentioned methods.

• In this section, for data quality assurance, please mention the Cronbach’s alpha for each questionnaire you have assessed.

• The authors stated that approximately one-sixth (16.6% or 84) of the sample were unable to read and write. How were the questionnaires completed for these individuals?

• The sample size was 521 individuals, and ultimately 506 were included. Please mention the reasons for excluding individuals.

• The authors stated that they used the PHQ to measure social support; please provide a reference that this has been done previously. A four-point Likert scale of PHQ-9 and three items from the Oslo social support scale were also used to assess depression and participants’ level of social support.

• In Table 4, based on which variables are the adjustments reported?

• Severe depression is also significant in Table 4, but it is not addressed in the discussion section.

Reviewer #2: Dear Dr. Baradaran,

Thank you very much for the opportunity to review this manuscript. The study addresses a significant issue, and I commend the authors for their efforts. Considering the importance and increasing prevalence of diabetes, it is necessary to investigate the mental disorders it can cause.

The authors of this manuscript utilized different questionnaires, including the diabetes distress scale (DDS17), PHQ-9, ASSIST, and Oslo 3 social support to assess the diabetes related distress, depression, substance abuse, and social support in a sample of diabetic patients in southern Ethiopia. They found significant determinants (e.g., physical activity, history of hypertension, social support, and complications).

However, after carefully evaluating this manuscript, I believe there are significant areas for improvement in terms of scientific rigor, statistical analysis, and overall presentation. Therefore, I must recommend a MAJOR revision for this manuscript.

That said, I have included some comments and suggestions that the authors may find helpful in refining their work. I hope these will assist them in improving the quality and impact of their research.

General Comments:

1. The text needs some grammatical editing.

2. Keywords: Please provide keywords from the MeSH database.

3. The authors mentioned that “all data are fully available without restriction” in their Data Availability statement, but I couldn’t find it in the review portal.

Title:

4. Please change “diabetic-related” to “diabetes-related”.

5. Clearly mention the study design (e.g., cross-sectional) in the title.

Abstract:

6. Abbreviations used in the text (e.g., aOR) must be written in complete form in their first use.

7. In the methods section, please mention the data collection tools (e.g., scales and questionnaires), study period, and eligibility criteria instead of the statistical software.

8. You can remove the software names from the methodology.

9. Please include the final analyzed sample size in the Results section.

10. Line 38: Please describe which complications.

Introduction:

11. Nearly all of the citations are irrelevant or from secondary sources. Please double-check all the citations and ensure they support the cited text.

12. Lines 73-79: I believe it is better to move this paragraph to the discussion section.

Methods:

13. Line 92: Please cite the CSA 2011 report.

14. Line 99: Please mention that this region is in Southern Ethiopia.

15. Population: How did you exclude the patients with a history of distress before the DM onset (to make sure that the distress is completely associated with the DM)? If not excluded, please mention this as a limitation of the study.

16. Line 104: How did you define the “critically ill”?

17. Lines 108-109: Please cite the previous study used for sample size calculation.

18. Lines 111-120: The text is somehow vague and hard to understand, and needs to be clearer.

19. Study variables: You can remove this section since important variables are discussed in the following text.

20. Again, citations in the Methods section are irrelevant or poorly support the cited text. Please double-check and make the necessary amendments.

21. Line 132: How was a patient considered to have DM? Which criteria were used? Please describe it clearly and in detail. If medical records were used, then please specify each hospital’s criteria.

22. Lines 147-148: Please cite the reference for the DDS-17 scale.

23. Line 169: Do you mean written informed consent by “written and signed permission letters”?

24. Line 173: Please change to “To minimize disruption to patient care, participants were approached only after their medical treatment was completed.”

Results:

25. Please consider reporting the frequency percents up to one decimal point.

26. Table 2: Based on the Methods section, patients were only included if they were diagnosed in the last 3 months. The information presented in the “Duration of living with diabetes” row is not consistent with the mentioned eligibility criteria.

27. Table 2: Please merge the “Diabetic associated complications” and “Type of complications” rows.

28. Table 2: Please mention the score/value range for each subcategory of the “Depression” and “BMI” rows.

29. Please edit the “bivariate” to “univariate” throughout your text.

30. Please edit the COR to simply OR and AOR to aOR.

31. Table 4: Please add a p-value column for the crude ORs.

32. Table 4: Please indicate the rows with OR or aOR equal to one are reference subcategories.

Discussion:

33. Please report the 95% confidence interval for the 11.46%.

34. Limitations: Please also mention that since all of the participants were selected from the hospital patients, this can lead to a selection bias because of including the patients with more severe DM.

Sincerely,

**Do you want your identity to be public for this peer review?** For information about this choice, including consent withdrawal, please see our Privacy Policy

Reviewer #1: No

Reviewer #2: No

---

## [Author Response · Author response to Decision Letter 1]

16 Aug 2025

1. more about the validity and reliability ( psychometry ) of employed questionnaires in your study in local language

Response : thank you for your comment. The reliability and validity of the questionnaire were incorporated under the 'data quality assurance' section.

2. This study needs mediation analysis for robust conclusion

Response:Thank you for your constructive feedback. We conducted a mediation analysis for this study and identified a variable that met the criteria to be considered a mediating variable. A detailed description of the analysis is provided in the 'Data Analysis' section, and the findings are reported in the 'Results' section of the study.

3.The authors wrote: “The sample size was determined by using the previous study variable of HbA1c (percentage of outcome in the unexposed group = 93.2% and adjusted odds ratio = 5.49).” Please add a reference to this section.

Response: Thank you for your comment. We have added a reference for the sample size at the end of the adjusted odds ratio for the study variable (HbA1c).

4. The authors stated that approximately one-sixth (16.6% or 84) of the sample were unable to read and write. How were the questionnaires completed for these individuals?

response: Thank you for your comment. This is the reason we decided to use two methods for administering the questionnaire: self-administered forms and face-to-face interviews. We applied a face-to-face interview to the patient who could not read or write.

5. Nearly all of the citations are irrelevant or from secondary sources. Please double-check all the citations and ensure they support the cited text.

Response: Thank you for your constructive comment. We replaced the secondary source citation with the primary (original) source.

6.The sample size was 521 individuals, and ultimately 506 were included. Please mention the reasons for excluding individuals.

Response:Thank you for raising this concern. These respondents were not intentionally excluded; they met the inclusion criteria, but, due to their busy schedules and being in a hurry to get to work, they were unwilling to complete the questionnaire and were therefore recorded as non-respondents. This reason is mentioned in the “limitations of the study” section

7. Please specify the type of diabetes in the title. At the end of the introduction, the authors mention “like type 1 diabetes mellitus,” but there is no reference to the type of diabetes in the methods section.

Response: Thank you for your comment. We revised the introduction to indicate that the study focuses on type 1 and type 2 diabetes mellitus. In addition, we stated at the operational definition that the study specifically focuses on these two types of diabetes.

8. Please explain the phrase “fulfilled the inclusion criteria.”

Response: The list inclusion criteria that need to be fulfilled were stated under a separate heading titled “inclusion and exclusion criteria” in the method part of the study.

9. Reference the validity and reliability of the questionnaires used in the target population.

Response: Thank you for your constructive comment. The validity and reliability (internal consistency) were addressed in the “Data quality assurance” part of the study.

---

## [Editor Report · Decision Letter 1]

20 Aug 2025

Diabetes related distress and associated factors among adult diabetes mallitus patients attending public hospital in Gedio zone, south Ethiopia: Mediation analysis

PONE-D-25-10454R1

Dear Dr. Mersha,

We’re pleased to inform you that your manuscript has been judged scientifically suitable for publication and will be formally accepted for publication once it meets all outstanding technical requirements.

Kind regards,

Hamid Reza Baradaran, M.D., Ph.D.,

Academic Editor

PLOS ONE
---

## [Editor Report · Acceptance letter]

PONE-D-25-10454R1

PLOS ONE

Dear Dr. Mersha,

I'm pleased to inform you that your manuscript has been deemed suitable for publication in PLOS ONE. Congratulations! Your manuscript is now being handed over to our production team.

Kind regards,

on behalf of

Professor Hamid Reza Baradaran

Academic Editor

PLOS ONE